# Volcanically hosted venting with indications of ultramafic influence at Aurora hydrothermal field on Gakkel Ridge

Christopher R. German ●[1] ✉, Eoghan P. Reeves ●[2], Andreas Türke ●[2,3,4], Alexander Diehl ●[3,4], Elmar Albers ●[4], Wolfgang Bach ●[3,4], Autun Purser ●[5], Sofia P. Ramalho ●[6], Stefano Suman[1], Christian Mertens ●[3,7], Maren Walter ●[3,7], Eva Ramirez-Llodra[8,9], Vera Schlindwein ●[3,5], Stefan Bünz ●[10] & Antje Boetius ●[3,5,11]

The Aurora hydrothermal system, Arctic Ocean, hosts active submarine venting within an extensive field of relict mineral deposits. Here we show the site is associated with a neovolcanic mound located within the Gakkel Ridge rift-valley floor, but deep-tow camera and sidescan surveys reveal the site to be ≥100 m across−unusually large for a volcanically hosted vent on a slow-spreading ridge and more comparable to tectonically hosted systems that require large time-integrated heat-fluxes to form. The hydrothermal plume emanating from Aurora exhibits much higher dissolved $CH_4/Mn$ values than typical basalt-hosted hydrothermal systems and, instead, closely resembles those of high-temperature ultramafic-influenced vents at slow-spreading ridges. We hypothesize that deep-penetrating fluid circulation may have sustained the prolonged venting evident at the Aurora hydrothermal field with a hydrothermal convection cell that can access ultramafic lithologies underlying anomalously thin ocean crust at this ultraslow spreading ridge setting. Our findings have implications for ultra-slow ridge cooling, global marine mineral distributions, and the diversity of geologic settings that can host abiotic organic synthesis - pertinent to the search for life beyond Earth.

Evidence for submarine venting has been detected in every ocean basin, along mid-ocean ridges of all spreading rates[1,2]. The Gakkel Ridge, which spans the ice-covered Arctic Ocean, is both the slowest-spreading mid-ocean ridge (0.7–1.4 cm/yr[3–5]) and the least accessible because of year-round ice-cover. As a consequence, it represents the last ocean basin on Earth where hydrothermal plume signals, diagnostic of submarine venting, have been tracked to their seafloor source[6]. Hydrothermal plume activity overlying the Gakkel Ridge was first detected in 2001 allowing 9 discrete sources of venting to be inferred[7,8]. One of these plume signatures was identified above the Aurora mound, where the westernmost Gakkel Ridge intersects the Lena Trough (Fig. 1a). The Aurora volcanic mound is ~3–4 km in

[1]Woods Hole Oceanographic Institution, Woods Hole, USA. [2]Department of Earth Science & Centre for Deep Sea Research, University of Bergen, Bergen, Norway. [3]MARUM—Center for Marine Environmental Sciences, University of Bremen, Bremen, Germany. [4]Faculty of Geosciences, University of Bremen, Bremen, Germany. [5]Alfred Wegener Institute Helmholtz Centre for Polar & Marine Research, Bremerhaven, Germany. [6]Centre for Environmental & Marine Studies (CESAM), Department of Biology, University of Aveiro, Aveiro, Portugal. [7]Institute of Environmental Physics, University of Bremen, Bremen, Germany. [8]Norwegian Institute for Water Research (NIVA), Oslo, Norway. [9]REV Ocean, Lysaker, Norway. [10]Centre for Arctic Gas Hydrate, Environment and Climate (CAGE), University of Tromso–The Arctic University of Norway, Tromsø, Norway. [11]Max Planck Institute for Marine Microbiology, Bremen, Germany. ✉e-mail: cgerman@whoi.edu

**Fig. 1 | Location map for the Aurora seamount, Gakkel Ridge. a** Location of the Aurora hydrothermal field (black star), together with eight additional hydrothermal plume sources (white stars) identified along the Gakkel Ridge during the 2001 Arctic Mid-Ocean Ridge Exploration (AMORE) expedition[7]. Figure made with Geo-MapApp (www.geomapapp.org) using data from the International Bathymetric Chart of the Arctic Ocean[69]. **b** Multibeam bathymetric map showing tracklines for three high resolution deep-tow surveys (OFOBS-04, −07 and −08) that passed directly over the Aurora vent site (black star). Colored lines denote tracklines for each deployment, chevrons indicate direction of travel. The multibeam bathymetry displayed here was collected from the RV *Kronprins Håkon* in 2019[16] and gridded at 15 m. Black rectangle delineates the area of sidescan sonar survey shown in Fig. 2.

diameter and ~150 m in height (≤3850 m water depth at its summit, Fig. 1b) and located toward the center of a second-order ridge segment which terminates, to the west, at the Lena Trough and is bounded to north and south by ~1000 m tall rift-valley walls. Hydrothermal plume activity within this deep rift-valley was first detected using an in situ optical backscatter instrument package, attached to a dredge that crossed the summit of the mound, revealing a mid-water particle-rich plume diagnostic of high-temperature submarine venting[7]. The contents of the dredge haul included hydrothermal sulfides and fresh glassy pillow basalts[7,8].

Since that 2001 expedition, no effort had been made to return to the Aurora vent-field, even as scientific advances in other ocean basins made more detailed investigations of venting along this ultra-slow ridge, beneath a permanently ice-covered ocean, ever more compelling. Continuing exploration of other ice-free ultraslow-spreading ridges (Southwest Indian Ridge, Mid-Cayman Rise, Mohns Ridge) has revealed an abundance and geologic diversity in styles of seafloor fluid flow that have expanded our understanding of the impact that submarine venting can have on marine mineral deposit formation[9], abiotic organic synthesis[10] and the potential for habitability and possibly even life on other ice-covered ocean worlds[11].

Here we report on two icebreaker cruises that returned to Aurora in 2014 and 2019. Our work has employed a combination of water-column biogeochemical analyses, seafloor (video and still) photography and deep-tow sidescan sonar. Water-column studies from our 2014 expedition have allowed us to characterize the biogeochemical nature of the Aurora hydrothermal plume, using a combination of He-isotope, Mn and $CH_4$ analyses while our 2019 seafloor investigations of the site, once located, included the use of multibeam bathymetry, deep-tow sidescan sonar surveys and co-registered camera surveys across the vent-field. Together, these approaches have allowed us to characterize the geologic setting of the Aurora site, the only

hydrothermal field to have been tracked to source anywhere in the Arctic Ocean to date, and to expand the known range of styles of seafloor venting, worldwide.

## Results and discussion

### Aurora vent-site location and seafloor characterization

The precise location of high-temperature venting at Aurora was first ascertained in 2014 guided by hydrothermal plume surveys using a Conductivity-Temperature-Depth (CTD)-rosette. That work was conducted in series with a set of 11 deep-tow camera surveys using the Alfred Wegener Institute (AWI)'s Ocean Floor Observing System (OFOS), across the summit of the Aurora mound. However, it was only during the final operation of that 2014 expedition that OFOS passed, briefly, directly over an active "black smoker" vent-source located ~500 m downslope from the summit to the south and west[6]. Equipped with that information, we returned to Aurora in 2019 and conducted a further series of deployments using an upgraded version of the Ocean Floor Observing & Bathymetry System (OFOBS) that was equipped with a high-resolution sidescan sonar instrument mounted in its tail-fin, together with its down-looking cameras (see Methods). OFOBS deployments were targeted to pass directly over the previously-identified vent-site, as best as could be achieved under prevailing ice-drift conditions (Fig. 1). Three deployments (OFOBS-04, −07 and −08) passed directly over the target (Fig. 1) and on one of those deployments (OFOBS-07) both the trajectory of the vehicle (course over ground) and the vehicle heading (orientation of the sonar) were sufficiently consistent for a mosaic of high-frequency sidescan sonar data to be extracted, crossing the target vent-site and acquiring sonar-images that extend out to a range of ~100 m across-track (Fig. 2a).

The first notable characteristic evident from these sonar data is that, away from the Aurora mound, much of the rift-valley floor is relatively smooth and flat (Fig. 2a). Photographic ground-truthing reveals these areas to be draped in >90% sediment cover (Fig. 2b) and a series of gravity cores collected from the 2014 study, beyond the limits of the sonar study area imaged here, confirm a cover of ~3.5–5.5 m of gray muddy clays across the rift-valley floor throughout the local area[6]. At the summit of the Aurora mound, by contrast, and along its steep-sided scarps (water depths ≤3850 m in Fig. 1), abundant outcrops of basaltic pillow lava are exposed, often colonized by glass sponge filter feeders[6]. Notably, however, the vents that have been located at Aurora are not situated at the summit of this volcanic mound. Instead, they are situated ~500 m downslope to the south (Fig. 1), at the summit of a separate steep-sided ridge that extends for ~400 m to the north and east, away from the site of active venting as far as ~82°54.0'N, 006°13.5'W. During each deployment, the OFOBS vehicle had to be raised sharply to climb and pass over this ridge before descending the other side. The strike and location of the spine of this ridge is readily discernable from the high-resolution sidescan data in Fig. 2a. The ridge strikes oblique to the direction of travel of the instrument and, because of the low altitude of the deep-tow vehicle, long shadows were recorded in the data first to starboard (SE) and then to port (NW) as the vehicle approached and then crossed the summit of this steep-sided ridge. When measured in the across-strike direction (NW-SE) the breadth of this ridge appears to be <50 m. Because the OFOBS vehicle had to be raised some 10-15 m to clear the top of this ridge, and then lowered rapidly by the same amount to keep the seafloor in view of the cameras, we can calculate that the slopes of the ridge-flanks must be steep (30-45°). In the sidescan sonar imagery, this ridge is distinctive because it is decorated with high-backscatter angular fragments along its spine and flanks. At least some of these features appear to be narrow upright structures that cast long sharp-sided shadows, resembling sulfide chimneys rather than volcanic flow morphologies. By design, the low altitudes at which our OFOBS surveys were conducted have allowed us to ground-truth our sidescan observations with direct geological observations using both video imagery and still

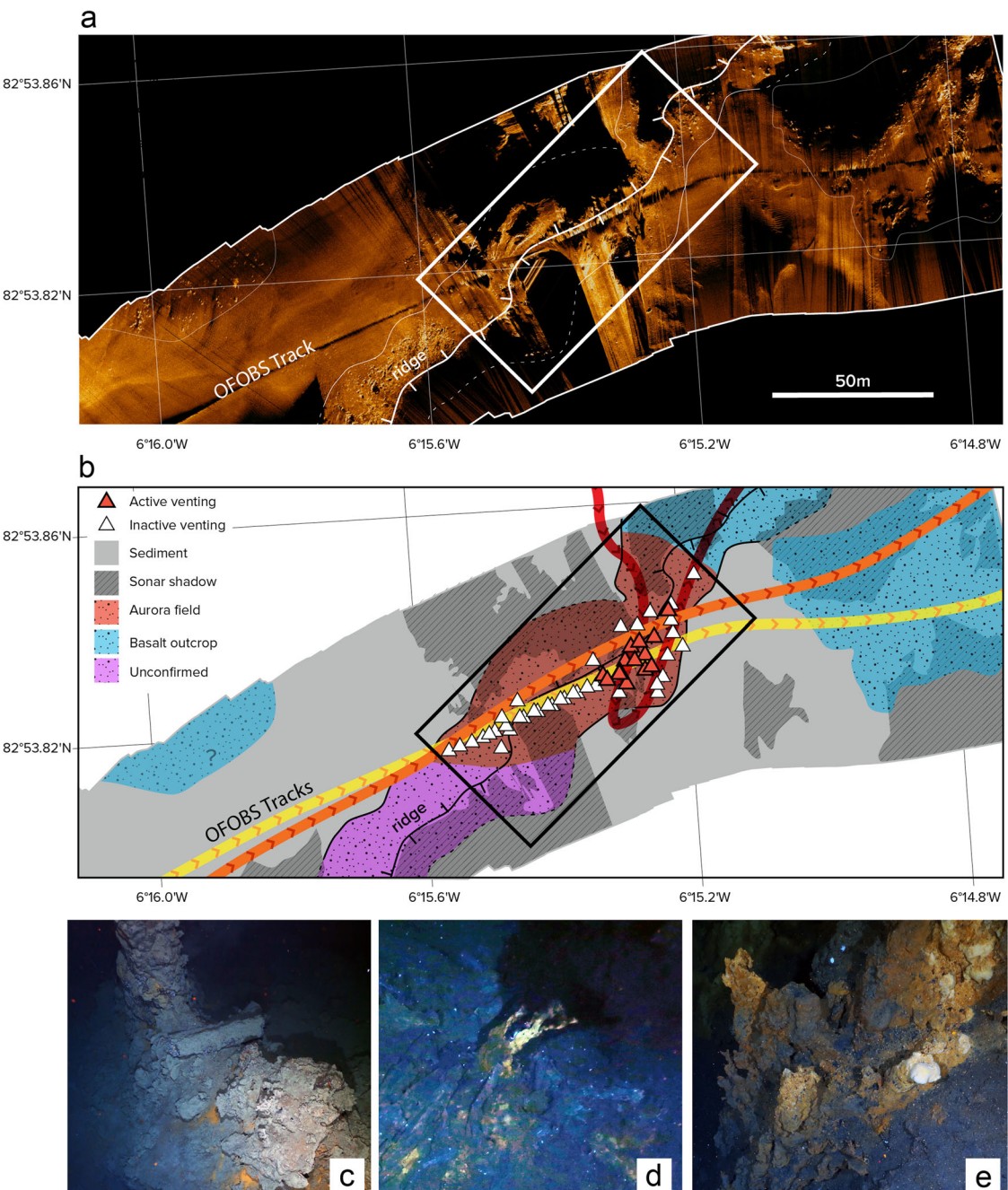

**Fig. 2 | Results from high resolution deep-tow (OFOBS) surveys of the seafloor at Aurora hydrothermal field. a** High-frequency sidescan sonar image of the seafloor crossing the Aurora vent-field, Gakkel Ridge from the OFOBS-07 survey. **b** Corresponding geological map based on sidescan interpretation coupled with deep-tow camera ground-truthing (dive tracks are in the same colors as Fig. 1, arrows show direction of travel). Observations of extinct sulfides are located as white triangles; observations of active fluid flow are located as red triangles. The ridge along which the vents are distributed is shown as a solid line with tick marks to indicate the downhill direction in (**a**) and (**b**) and the rectangle bracketing the confirmed hydrothermal field in both panels measures 50 m x 100 m. Note, however, that the southern limit to the Aurora field, along-ridge, remains unconstrained by camera ground-truthing. **c** Relict sulfides could be identified in photographs in multiple forms, all present in this image (from top left to lower right): upright, extinct chimneys; fallen sub-cylindrical chimney structures; and massive sulfide, often with distinctive red-orange oxidation weathering. **d** Active vent-sites could occasionally be identified from oblique photographs, as in this example, or more commonly from buoyant plumes of black billowing smoke that engulfed the cameras when the instrument passed more directly over a vent-source. **e** Schlieren effects from shimmering water together with pale, smooth-textured patches allowed us to identify lower-temperature hydrothermal flow and associated (putative) microbial mats. Scale bars in panels **c**, **d**, **e**: 1 m.

photographs – co-registered with the sidescan data collected during the OFOBS 07 deployment (where photographs are aligned along the nadir of the sidescan track, Fig. 2a). This combined approach has allowed us to confirm the presence of both active and inactive venting within some of the high-backscatter regions identified in Fig. 2a, and also to confirm where other high-backscatter textures indicate

basalt outcrops among sediment with no hydrothermal characteristics (Fig. 2b).

From still images collected by OFOBS, we have been able to identify areas of seafloor characterized by extinct venting, in the form of upright and collapsed chimneys, and as massive sulfide which is typically accompanied by rust-orange colored weathering (Fig. 2c).

Wherever such features have been identified, their locations are shown as white triangles in Fig. 2b. Also readily discernible, both in still images and in the live video-feed, are the locations where vigorous, presumably high temperature, "black smoker" venting was identified—either from images of active chimneys themselves (Fig. 2d), or from images showing that OFOBS' down-looking cameras were engulfed in buoyant plumes of black smoke immediately overlying active vents. Particulate metal-sulfide "smoke" formation in seafloor "black smoker" fluids is primarily driven by precipitation of abundant dissolved iron (Fe[12]). Like many chalcophile metals, Fe mobility and abundance in hydrothermal fluids is strongly dependent on temperature, and increases dramatically (logarithmically) above ~300 °C[13,14]. The vigorous "black smoker" venting observed by OFOBS therefore is entirely consistent with high-temperature (≥300 °C) venting at Aurora. Further, our OFOBS observations reveal that this active high-temperature venting is concentrated along the spine of the ridge (Fig. 2a, b), with areas of inferred lower-temperature flow evident at its periphery. This "diffuse" venting (visible as clear, unfocused fluids lacking particulates and thus likely much lower in temperature) is not as obvious to detect as high temperature "black-smoker" flow in still images (Fig. 2e), but can be discerned readily from co-registered video recordings where the schlieren effect from shimmering warm water is particularly apparent. Even in still images, however, regions of lower-temperature flow at Aurora exhibit what appear to be diagnostic visual cues: a distinctive pale cream-yellow coloration and an anomalously smooth-textured morphology that, at the comparably deep, ultra-slow spreading Mid-Cayman Rise, have been demonstrated to be microbial in origin[15]. All sites of active fluid flow at Aurora, whether due to diffuse or high-temperature venting, are demarcated by red triangles in Fig. 2b.

## Hydrothermal plume characteristics

In 2014, a total of 11 CTD "floe-yo" casts were conducted that passed across or close by the summit of the Aurora mound (Fig. 3a). During each of these casts, the CTD-rosette was raised and lowered through the deep-water column as the ship drifted with ice-movement. Further CTD casts were conducted in 2019[16] but those were targeted at the periphery of the Aurora non-buoyant plume[17] and are not included in the results discussed here. In 2014, our initial search strategy was focused on the shallowest summit of the Aurora mound, but buoyant plume signals were subsequently detected ~500 m downslope to the southwest, presaging first imaging of active venting on the final deployment of that cruise[6]. The cumulative depth-profiles for water samples collected across all of our 2014 CTD casts are shown (Fig. 3b–d) for δ³He anomalies and for dissolved $CH_4$ and dissolved Mn concentrations. These profiles all show enrichments at depths between 2800 m and 3700 m, consistent with a single coherent non-buoyant plume dispersing away from a source of high-temperature "black smoker" venting on the underlying seafloor at ≥3800 m depth. The maximum δ³He values reported at plume-height at Aurora (up to 14.7%) are entirely consistent with a mantle origin / hydrothermal source, while the high concentrations of $CH_4$ and Mn (up to 30 nM) observed at the same depths are consistent with hydrothermal plume values observed within ≤5 km of other active high-temperature vents[12].

## Laterally extensive venting in a neovolcanic setting

Previously, it had been reported that the Aurora hydrothermal field is associated with a small volcanic cone characterized by fresh glassy pillow basalts, deep within the rift-valley, toward the center of a second-order ridge segment at the very western end of the Gakkel Ridge[7,8]. Because of the volcanic morphology of the site, and because it was located within the region of the Gakkel Ridge (everywhere west of 3°E) that has been classified as the Western Volcanic Zone, the geologic controls on venting at Aurora had been assumed to correspond to similarly small, volcanically hosted vent systems typified by the Snake

Pit and Broken Spur sites on the slow-spreading Mid-Atlantic Ridge[18,19]. In such settings, it has been calculated that modest (~100 MW) hydrothermal venting can suffice to extract all of the heat associated with emplacement of young ocean crust through "quantum events" of dike intrusion over decadal time-scales[20]. By contrast, construction of larger vent-deposits, such as the tectonically hosted TAG and Rainbow hydrothermal fields, appear to require much larger time-integrated heat fluxes focused at a single site, requiring energy equivalent to cooling to the base of the ocean crust for ≥10 km along axis over timescales of ~1–10 kyr[21]. At Aurora, the lateral extent of the hydrothermal field revealed by our direct seafloor observations suggests that it falls into this latter category of sustained venting. Present-day active fluid flow extends for ~30 m in the along-track direction and cumulative sulfide deposition extends for ~100 m. These estimates may prove conservative. In the deep-tow sidescan data, it is apparent that our ground-truthed observations of sulfide deposition and active venting are directly associated with a steep-sided ridge, decorated with high back-scatter angular features (Fig. 2). To the north, upslope toward the Aurora mound, our camera observations allow us to map the contact between the hydrothermal field and basaltic outcrops, but to the south, where similar acoustic textures persist along the spine of the ridge, the limits to the hydrothermal field remain unconstrained. Comparably extensive hydrothermal deposits have been reported previously from other neovolcanic-hosted vent sites on ultra-slow spreading ridges: at Loki's Castle on the Mohns Ridge[22], the Dragon site on the SW Indian Ridge[23], and Piccard on the Mid-Cayman Rise[24]. In all of these cases, it has been argued that the generation of substantive sulfide deposits *requires* long-lived focused flow even though they manifest no evidence for tectonic control. The processes required to sustain such systems, therefore, remain to be resolved[9].

## Evidence for subsurface ultramafic influence at Aurora

Our hydrothermal plume data may provide further important insights into subsurface processes at Aurora, beyond what can be resolved from purely surficial observations and dredging. While the vertical profiles for each of the tracers, δ³He, $CH_4$ and Mn, in isolation, exhibit hydrothermal plume-profiles characteristic of high-temperature "black-smoker" venting (Fig. 3), in agreement with visual OFOBS observations, the high $CH_4$:Mn ratio (~1:1) for the entire sample suite defines a common trend that is not consistent with basaltic/gabbroic influences, alone, on the composition of fluids exiting the seafloor (Fig. 4). Drawing from a recent compilation of all vent-fluid data reported from submarine hydrothermal sites worldwide[25,26], the majority of high temperature (≥300 °C) hydrothermal fields situated in basaltic/gabbroic rocks generally exhibit low $CH_4$:Mn ratios (≤1:3). These ratios are largely driven by the low $CH_4$ abundances typical of these systems, which may be driven either by a lack of $H_2$ to promote extensive carbon reduction to form substantial abiotic $CH_4$, or perhaps geologic controls that limit extraction of such pre-formed abiotic $CH_4$ from mineral-hosted inclusions[27]. The only exception to that rule is at Lucky Strike ($CH_4$/Mn = 3.8 ± 0.9) which is highly anomalous geologically, as well as geochemically. Situated at 37°15'N, on the MAR, the Lucky Strike segment hosts an unusually tall (>500 m) conical seamount that fills the segment center and reaches heights comparable to its rift-valley walls. No other ridge-segment shows matching morphological features within the 15-40°N MAR survey area[28], nor does any such feature occur anywhere along the Gakkel Ridge[8]. Construction of this unusually large conical volcanic edifice has been attributed to efficient melt focusing coupled with high effusion rates[29]. The high vent-fluid $CH_4$:Mn ratios at Lucky Strike (which are largely due to anomalously high $CH_4$ rather than unusually low Mn) have been attributed to a particularly deep-rooted convection cell beneath this unusually large seamount, which may more extensively 'mine' dissolved $CH_4$ from igneous fluid inclusions at depth[27]. Alternately, the high $CH_4$ at Lucky Strike has also been tentatively attributed to

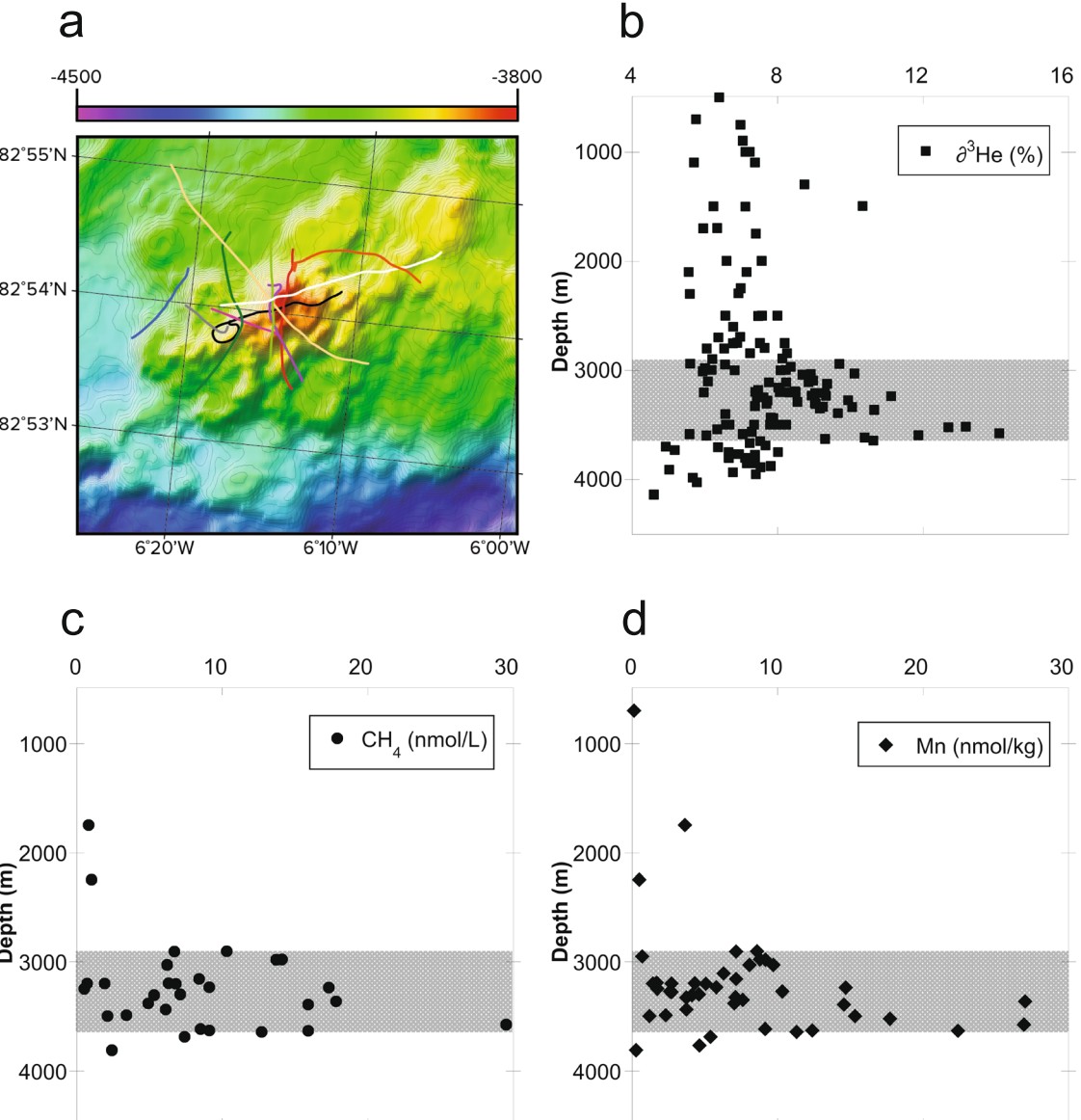

**Fig. 3 | Plots of Aurora hydrothermal plume composition. a** Location map for the trajectories of 11 CTD deployments over the Aurora mound, completed in 2014[6] projected over multibeam data from 2019[16] that is gridded at 20 m. Each CTD deployment was selected based on interpretation and prediction of ice-floe trajectories that dictated the direction of travel. Cumulative depth profiles are shown for **b** δ³He anomalies, **c** dissolved $CH_4$ concentrations and **d** dissolved Mn concentrations, as measured in water samples collected during the CTD casts. Shading illustrates the depth range (2800–3700 m) of the non-buoyant hydrothermal plume.

potential influence from the carbon-enriched Azores mantle plume[30]. Two shallower (<1 km depth) Azores systems, Bubbylon (37°48'N) and Menez Gwen (37°50'N) can also exhibit high $CH_4$:Mn plume ratios (≤20.7[31]) but the extremely high (milli-molar) $CH_4$ vent-fluid concentrations at those sites can be attributed to vigorous fluid boiling and widespread gas-bubble exsolution at the seafloor[32]. Because Aurora lacks any morphological evidence for sustained melt focusing/lava effusion or proximal hot-spot effects and is far too deep for seafloor fluid boiling to occur, we do not consider any of these Azores hot-spot systems to be pertinent analogs to explain the data reported here, even though Aurora is situated on a slow-spreading ridge and hosts high $CH_4$/Mn hydrothermal emissions in close proximity to seafloor basaltic outcrops.

Rather, as hydrothermal exploration along slow and ultra-slow spreading ridges has progressed, a series of "black smoker" systems have been found that exhibit evidence for ultra-mafic influence in their endmember vent-fluid compositions in the form of: high dissolved Cu

concentrations, high dissolved $H_2$ concentrations and high endmember vent-fluid $CH_4$:Mn ratios relative to basalt-hosted systems[33]. The latter reflects the increased contributions of abiotic $CH_4$ to fluids relative to Mn[27]. The best characterized examples of vents that meet this description are five sites along the Mid-Atlantic Ridge (Ashadze 1, Ashadze 2, Logatchev 1, Logatchev 2 and Rainbow) which, together, define a range in vent-fluid $CH_4$:Mn ratios of 0.47-7.1,bracketing the plume composition for Aurora (Fig. 4). Other ultramafic-hosted fluid-flow sites on slow- and ultra-slow spreading ridges have also been reported—from the Mid-Atlantic Ridge (Lost City[34]); the Mid-Cayman Rise (Von Damm[10]) and the SW Indian Ridge (Old City[35]). Those systems, however, represent poor geological analogs because they host much lower-temperature (<226 °C) clear fluid venting rather than the vigorous (and presumptively much higher temperature) "black smoker" vents observed at Aurora. Those lower-temperature sites also exhibit $CH_4$:Mn ratios >30:1[25,26] primarily due to a lack of substantial Mn which, similar to Fe, is only extensively mobilized during high

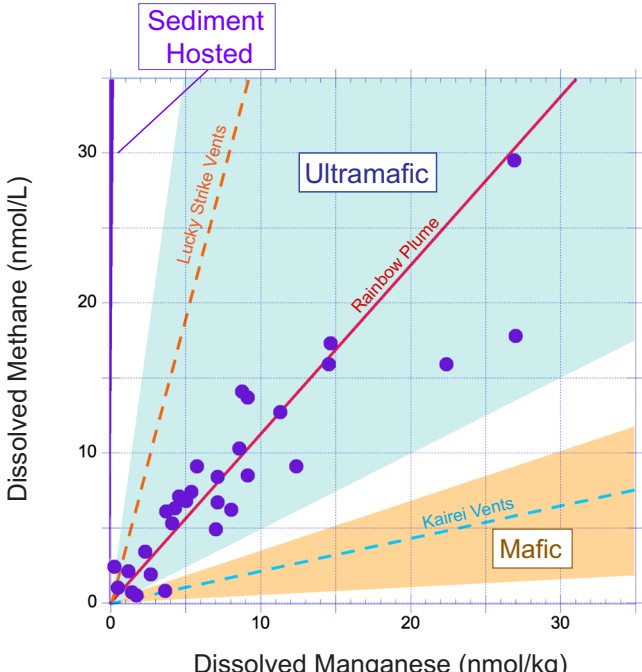

**Fig. 4 | Plot of dissolved CH4 concentrations vs dissolved Mn concentrations for the Aurora hydrothermal plume (solid circles) compared with the like-for-like trend (red line) for the Rainbow hydrothermal plume**[39]. Ranges of CH4:Mn ratios for high temperature (≥300 °C) vent-fluids at sites hosted in mafic rocks on slow and ultra-slow spreading ridges (Snakepit, TAG, Broken Spur and Piccard) are shown in pale orange (data from[59,70,71]). Corresponding ranges for high temperature (≥300 °C) vents influenced by ultramafic rocks (Rainbow, Logatchev I, Logatchev II, Ashadze I, Ashadze II) are shown in pale blue (data from[33,71,72]). Also shown, in purple, are the much higher CH4:Mn values for systems influenced by terrigenous continental sediments of the kind reported at Aurora (ref. [40] and reference. therein). The dashed orange line (Lucky Strike Vents[73]) and dashed blue line (Kairei Vents[36]) indicate two vent-systems that depart from the general case, each of which is discussed in detail in the text.

temperature (>300 °C) alteration[13,14]. One other hydrothermal system that has been reported to exhibit evidence for ultramafic influence (albeit in lithologies that are not typical of mantle peridotites) is the Kairei hydrothermal field, Central Indian Ridge. There black smoker vent-fluids exhibit anomalous enrichments in dissolved gas concentrations ($H_2$: 2.5-3.7 mM; $CH_4$: 0.12-0.20 mM[36]) that, nevertheless, are significantly lower than their Mid-Atlantic Ridge counterparts ($H_2$: 11-19 mM; $CH_4$: 0.8–2.3 mM[33]) resulting in a distinctly lower CH4:Mn ratio (Fig. 4). However, Kairei is considered to be distinct from all Mid-Atlantic Ridge (MAR) ultramafic systems and, instead, to represent its own class of magma-assisted, ultramafic hosted, hydrogen-rich hydrothermal system[37]. Given its distinctly unusual characteristics, with properties that cannot explain the high CH4:Mn ratio of the Aurora plume, we do not consider Kairei further, here. Rather, we note that the anomalies we observe from the Aurora plume, on the ultra-slow spreading Gakkel Ridge, fall within the range of all known high-temperature vent-fluids from ultramafic-influenced "black smoker" vents on the Mid-Atlantic Ridge which, as a slow-spreading ridge, provides a closer point of comparison for this work.

Of course, it is important to remember that the data that we report here are from a hydrothermal plume located hundreds of meters up in the water column above the seafloor, rather than from end-member vent-fluids. It has been well recognized, elsewhere, that both dissolved methane and dissolved Mn can behave non-conservatively in dispersing plumes due to both abiotic and micro-biologically mediated processes[12,38]. Over short time/length-scales of dispersion, however, these same tracers can behave quasi-

conservatively due to slow oxidation kinetics of both Mn and $CH_4$ relative to plume transport rates[39]. Because all of our samples were collected within ≤3 km of the identified vent-source at Aurora (Fig. 3a), and because the resultant data define a single trend (Fig. 4) with no marked evidence of fractionation, we infer that the ratio measured in the Aurora plume should be very close to its source vent-fluids.

Informed by analyses from the nearest-known vent-site to Aurora, Loki's Castle to the far south[22], one final class of hydrothermal field is considered, here: sediment-hosted hydrothermal systems which exhibit CH4:Mn ratios 1–2 orders of magnitude greater than Aurora[25,26]. At Loki's Castle, located at the intersection of the ultraslow spreading Mohns and Knipovich Ridges, end-member vent-fluids exhibit high CH4:Mn ratios that fall in the range 170-245[40]. Previously, experimental studies reacting terrigenous sediment with seawater-derived hydrothermal fluid have provided evidence that such distinct high CH4:Mn ratios arise from a profuse generation of dissolved thermogenic $CH_4$, derived from sedimentary organic matter, that is only accompanied by release of dissolved Mn at concentrations similar to those reported from basalt and ultramafic rock alteration experiments[41–43]. Two Pacific vent-sites where the mid-ocean ridge is covered with terrigenous sediments deposited from the adjacent continent[44,45] exhibit CH4:Mn ratios that overlap with both these experimentally produced ratios[41–43] and the end-member vent fluid ratios from Loki's Castle. At Loki's Castle, therefore, high vent-fluid CH4:Mn ratios have been attributed to deep-seated hydrothermal alteration of sediments occluded in the magmatic-hydrothermal system beneath the Mohns Ridge, much as has also been invoked previously for the Main Endeavor Field on the Juan de Fuca Ridge[46]. Given the vastly dissimilar nature of the CH4:Mn ratios between Loki's Castle and Aurora, however, coupled to the absence of any thick sediment cover at Gakkel Ridge (≤5.5 m observed at the seafloor[6]) we do not consider it reasonable to infer any significant influence of hydrothermal sediment alteration to the vent fluids at Aurora. Instead, we hypothesize that the moderate CH4:Mn ratios that we have observed, in concert with the lateral extent of the vent-field we have mapped, are much more likely to result from the interaction of high-temperature vent-fluids with ultramafic lithologies beneath the seafloor.

As discussed at the start of this section, ultramafic-influenced hydrothermal vents on slow spreading ridges are unusually enriched in dissolved hydrogen as well as dissolved $CH_4$ arising from the serpentinization of mantle peridotites[33]. By contrast, $H_2$ is not commonly measured in hydrothermal plumes overlying basalt-hosted systems due to its highly labile nature which means that it is sparingly detectable in hydrothermal plumes, even in samples collected from directly above a known vent[38]. Because all prior work at Aurora had indicated that the site should be neo-volcanically hosted, however[7,8] we did not plan a priori for at-sea measurements of dissolved $H_2$[6]. Serendipitously, however, a complementary study of the Aurora plume's microbiology has provided independent, albeit indirect, evidence for dissolved hydrogen enrichment in the Aurora hydrothermal plume[47]. That work reports the isolation of a new species of microbe that dominates the microbial communities in the Aurora plume and exhibits traits indicative of an obligate adaptation to hydrogen-rich hydrothermal plumes. Additionally, that same study has identified that bacterial hydrogenase enzymes are highly expressed within the plume, but not outside of it, implying that the Aurora system may well be enriched in dissolved $H_2$ as well as dissolved $CH_4$. Although this is by no means definitive, it nonetheless provides an additional line of evidence that would be consistent with the Aurora hydrothermal convection cell interacting with ultramafic lithologies beneath the seafloor.

Such a finding need not be controversial. Along all ultra-slow spreading ridges, conductive cooling limits mantle melting[48] and melt production can be highly discontinuous[49]. At one extreme, ultraslow ridges can exhume mantle rocks in the complete absence of any ocean crust lithologies over millions of years[50]. The Aurora site is located at

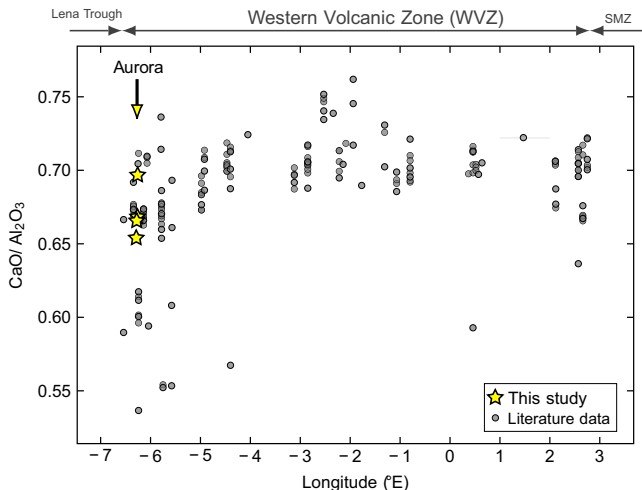

**Fig. 5 | Changes in basalt CaO/Al2O3 compositions along the Western Volcanic Zone (WVZ) of the Gakkel Ridge between the Lena Trough and the Sparsely Magmatic Zone (SMZ).** Lower ratios towards either end of the WVZ compared to its center can be attributed to (i) lower degrees of partial melting of the mantle causing less clinopyroxene to contribute CaO to the melt and resulting in reduced melt production[51], and/or (ii) high-pressure crystallization of clinopyroxene, leading to CaO depletion in the melt, within a thick lithosphere[52]. In either case, the data support a restricted thickness of volcanic crust overlying the lithospheric mantle at the Aurora site in the western WVZ (location denoted by vertical arrow). Yellow stars: this study. Gray circles: glass analyses downloaded from the PetDB Petrology DataBase (earthchem.org/petdb).

the westernmost limit of the Gakkel Ridge, where the Western Volcanic Zone (WVZ) intersects the Lena Trough (Fig. 1). The WVZ, in general, exhibits subdued magmatic segmentation in which numerous small volcanic cones are bound by stepped rift-valley walls and short-throw faults[8]. Detailed investigation of basalt data along the WVZ, however, reveal evidence for anomalous behavior in the Aurora segment, closest to the Lena Trough. Specifically, the geochemical signature of low extents of melting in mid-ocean ridge basalts are relatively high contents of $Na_2O$ and low $CaO/Al_2O_3$[51]. When viewed through that lens, basalt compositions along the WVZ reveal evidence for diminishing degrees of partial melting and/or increasing extents of high-pressure fractional crystallization of clinopyroxene[52] towards the westernmost end of the WVZ where the Aurora ridge segment is located (Fig. 5). This argues for a reduced thickness of basaltic crust with the possibility of an increased lithospheric thickness beneath the Aurora site. Such a reduced magma supply at the WVZ's western limit would be entirely consistent with the nearby intersection with the Lena Trough (Fig. 1a). The latter is considered nonvolcanic and has ultramafic rocks commonly exposed at the seafloor[53]. At Aurora, therefore, the pillow basalts we have observed in outcrop at the seafloor may only represent a relatively thin veneer of oceanic crust while the hydrothermal convection cell accesses ultramafic lithologies beneath.

### Future work and broader implications
Future geophysical surveys of the Aurora hydrothermal field could investigate the crustal and lithospheric structure of this section of the ridge-axis. Previously, seismic profiling has confirmed that the ocean crust farther east along the Gakkel Ridge *is* anomalously thin, at just 1.9-3.3 km[54], just as has been reported on a like-for-like basis from the South West Indian Ridge (SWIR: ≥ 2.5 km[55]) but beneath the Western Volcanic Zone, the lithospheric structure remains poorly constrained. In parallel, detailed geochemical analyses of endmember vent fluids at Aurora should in future allow the nature of the subsurface circulation to be constrained. If our hypothesis is correct, the work reported here has important implications for multiple fields.

Ultraslow-spreading ridges represent ~25% of the cumulative length of the global ridge crest[56] but were initially overlooked with respect to hydrothermal circulation because of an assumption that hydrothermal cooling was linked directly to spreading rate[57]. Even following the first discoveries of evidence for venting along the SWIR and the Gakkel Ridge, however[7,58], their very inaccessibility meant that detailed seafloor investigations only followed more than a decade later ([23], *this study*). Along the much more accessible Mid-Cayman Rise, however, the two new hydrothermal fields that have been located, Von Damm and Piccard, have extended the range of all previously-known hydrothermal systems[10,59]. From that perspective, therefore, perhaps we should not be surprised that the Aurora vent-site also appears to have extended that known range still further. Rather, we should expect that continued exploration of ultraslow ridges will continue to expand our understanding of both the nature of crust-mantle interactions that can arise and the forms of hydrothermal cooling that they can host. From an economic geology perspective, we have already noted that much larger hydrothermal fields can arise on ultraslow spreading ridges than had previously been anticipated for volcanic-hosted systems, where—once again—Lucky Strike has proven anomalous[9]. Along the slow-spreading Mid-Atlantic Ridge, all other volcanically hosted systems generate small, relatively copper-poor and gold-poor deposits whereas tectonically hosted, long-lived systems generate larger metal-rich deposits with much greater economic potential. If large volcanically hosted systems such as Aurora do, indeed, prove to be ultramafic influenced at depth, the seafloor mineral deposits they produce may be large and enriched in copper and gold, just like their tectonically hosted counterparts.

Equally societally relevant, but from a quite distinct perspective, the discovery of ultramafic-influenced venting beneath the permanent ice-cover of the Arctic Ocean could have important implications for forthcoming space missions to explore for life beyond Earth. It has been hypothesized that ancient seafloor hydrothermal systems, with reducing $H_2$-rich fluids, could have facilitated the abiotic synthesis of key prebiotic organic molecules from inorganic carbon and, hence, have played an important role in the origin of life[60]. Of course, the primary competing "Warm Pond" hypothesis for the origin of life on Earth is much longer-established[61] but the submarine vent hypothesis has received significant recent impetus with our demonstration that de novo abiotic organic synthesis can and does occur in a modern-day ultramafic-hosted, $H_2$-rich hydrothermal system at the Von Damm hydrothermal field on the Mid-Cayman Rise[10]. This is important because contemporaneous work from the US National Aeronautic and Space Administration (NASA)'s Cassini mission has revealed evidence for submarine venting on Saturn's moon Enceladus[62] and, further, that high concentrations of $H_2$ and $CH_4$ may be present in Enceladus' plumes because its seafloor vents are also ultramafic-hosted[11]. Taken together, these discoveries are profound—they imply that the oceans on Enceladus could be habitable for microbial life and, potentially, inhabited[63]. The recently-released National Academies' report *Origins, Worlds and Life*[64] includes a specific recommendation that NASA should initiate a new mission to search for life on Enceladus within the coming decade. If our hypothesis is correct, and Aurora is an ultramafic-influenced system similar to Von Damm, this $CH_4$-rich (and, potentially, $H_2$-rich) hydrothermal field at the base of an ice-covered ocean, could provide an ideal natural laboratory for future research that helps guide such a civilization-impacting endeavor.

## Methods
### Field campaigns
Fieldwork for this research was conducted aboard RV *Polarstern* cruise PS86 (PIs A.Boetius, W.Bach & V.Schlindwein) in 2014[6] and RV *Kronprins Haakon* cruise 2019708 (PIs S.Bünz & E.Ramirez-Llodra) in 2019[16]. Data presented here include water column plume data collected from a CTD-rosette at multiple locations across the summit of the Aurora

seamount in 2014 and from sidescan sonar and seafloor photography collected using the Alfred Wegener Institute, Helmholtz Centre for Polar and Marine Research (AWI)'s OFOBS deep-tow survey package in 2019.

## Hydrothermal plume sampling and analyses

Water-column samples discussed here were collected in 2014 using a CTD-rosette equipped with a Seabird 9/11+ CTD rosette carrying in situ particle and redox sensors specially selected for the detection of hydrothermal plumes. Casts were planned to pass as close as possible to the summit of the Aurora volcanic mound, using modeled predictions for the trajectory of ice-floes that dictated the motion of the ship. The strategy was to lower and raise the CTD-rosette through the deep-water column along the path of each deployment to bracket the depth range across which in situ redox and particle anomalies were apparent (typically between 2800 m and 3700 m water-depth). Tracks for the CTD-package relative to the seafloor were monitored in real-time using an ultra-short baseline (USBL) navigation system with a beacon mounted on the rosette. Samples were collected using Niskin bottles with silicone internal fittings that were mounted on the CTD rosette. Up to 22 samples were collected from each cast across a range of depths spanning the hydrothermal plume as detected using in situ sensor data. Upon recovery on deck, water samples were drawn for a combination of ship-board and shore-based laboratory analyses for three key hydrothermal tracers: $\delta^3He$, dissolved methane ($CH_4$) and dissolved manganese (Mn). Samples for helium isotope analyses were drawn into previously evacuated glass ampoules, direct from the Niskin bottles and then flame sealed for return to the laboratory[65]. These samples were then analyzed ashore in the Institute of Environmental Physics in Bremen[66]. Samples for dissolved $CH_4$ were drawn from the same Niskin bottles and analyzed at sea by headspace extraction gas chromatography. Water samples for dissolved Mn were collected from the same bottles as the helium and methane samples and filtered through 0.45 μm cellulose membrane filters into pre-washed acid-cleaned bottles then acidified using quartz-distilled 2 N $HNO_3$ prior to return to the laboratory in Bremen. Thereafter all hydrothermal metal analyses were conducted by Inductively Coupled Plasma Mass Spectrometry.

## Deep-tow camera and sidescan sonar surveys

In 2014, hydrothermal venting was located from a single pass over the vent-site during the final deployment of the cruise, using AWI's *OFOS* deep-tow package[6]. (The 2014 OFOS tracks across Aurora's summit can be viewed at http://doi.pangaea.de/10.1594/PANGAEA.836641). In 2019, four deployments of AWI's upgraded *OFOBS* package[67,68] were used to target the hydrothermal site using ultra-short baseline (USBL) navigation. Operating from a deep-tow conducting cable with a fiber-optic core, OFOBS hosts an HDTV camera that provides real-time video of the seafloor as surveys are undertaken, together with an 8MPix digital still camera that can be programmed to take images at pre-set time intervals throughout a deployment and that can also be operator-instructed to take additional photographs on demand in response to features detected in real-time via the video feed. On this cruise, the pre-set interval between still images was 20 s. At a nominal altitude of 5 m, the camera systems provide a field of view ~4 m × 3 m on a side (with fixed angular geometry, lowering the camera to the seafloor to obtain higher resolution imagery—e.g. to aid with biological species identification—reduces the size of the area that is imaged). In addition, to the photography that can be conducted using *OFOBS*, a tail section to the system houses a high-frequency sidescan sonar instrument mounted between vanes designed to orient the vehicle in the along-track direction. While down-looking photography only allows a narrow strip of the seafloor to be imaged during an *OFOBS* tow, this co-registered sidescan insonification can provide a broader context for that imagery out to ~100 m in the across-track direction (50 m to each side) at

camera survey altitude. All video and still imagery were collected and recorded aboard ship in real-time as the surveys progressed. Sidescan sonar data from this cruise was processed post-cruise in the Deep Submergence Laboratory at WHOI. First, sidescan sonar data was merged with 1 Hz navigation from the USBL beacon attached to OFOBS using *Chesapeake NavInjector*© software. Next, *Chesapeake SonarWiz*© software was used to adjust gain on the data using Empirical Gain Normalization (EGN). Finally, individual sonar files were exported into a GeoTif file at a resolution of 0.1 m per pixel.

## Basalt analyses

Basalts were sampled during PS86[6]. Bulk rock compositions were analyzed on fused glass beads at the Institute for Chemistry and Biology of the Marine Environment, University of Oldenburg. Sample powder was mixed with $Li_2B_4O_7$ and $NH_4NO_3$ and heated to 500 °C (oxidation) before melting at 1450 °C and quenching into a fused glass bead.

Major and trace element concentrations were measured with a wavelength dispersive PANalytical Axios plus X-ray fluorescence spectrometer. Analyses of in-house reference materials were used to monitor the data quality: relative standard deviations as a measure of analytical accuracy are better than 2% for major elements, except for $K_2O$ and $P_2O_5$ (better than 5%), and better than 8% for trace elements.

## Data availability

All data collected for this study aboard FS Polarstern in 2014[6] have been deposited in the PANGAEA database and are available via data-links associated with the cruise report at https://www.pangaea.de/?q=PS86. The seafloor survey data collected from the OFOBS deep-tow instrument aboard FS Kronprins Haakon in 2019 for geological field mapping at Aurora[66] have also been uploaded to the Pangaea system at https://doi.org/10.1594/PANGAEA.943364. All other supporting data from the 2019 cruise are available via datalinks associated with the HACON cruise report at https://haconfrinatek.com/results/.

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

## Acknowledgements

We thank the officers and crew of FS Polarstern (PS86) and FS Kronprins Haakon (KPH708) especially U. Hoge (AWI) and the Nereid Under Ice team (WHOI) who assisted with OFOBS deployments. This research was supported by: NOAA Grants NA14OAR4320158 and NA19OAR0110406 (C.R.G., S.S.); NASA Grants NNX16AL04G and NSSC19K1427 (C.R.G., S.S.); the Helmholtz Association (A.B., V.S.); DFG Grant 49926684 (A.B., W.B.); the Max Planck Society, ERC Grant 294757 and AWI's FRAM program (A.B.); FRINATEK Grants 274330 (E.R.-L., E.P.R., S.B. and C.R.G.) & 223259 (S.B.); Shiptime support from UiT (S.B.); CESAM Contract CEE-CIND/00758/2017 (S.P.R.) and CESAM Grants UIDP/50017/2020, UIDB/50017/2020 & LA/P/0094/2020 (S.P.R.). M.Tominaga and N.Renier (WHOI) helped with preparation of the figures.

## Author contributions

C.R.G. (US), A.B. and V.S. (Germany) and E.R.-L. and S.B. (Norway) led this international collaboration. C.R.G., A.T., A.D., S.S., C.M., V.S. and A.B. conducted the field work in 2014 and C.R.G., E.P.R., A.P., S.P.R, S.S., E.R.-L. and S.B. conducted the field work in 2019. C.M. and M.W. were responsible for the CTD program at sea and noble gas analyses and all other geochemical analyses were undertaken by A.T., A.D., E.A. and W.B. C.R.G., E.P.R., A.T., A.D., E.A. and W.B. interpreted the geochemical data, C.R.G. and A.B. analyzed the 2014 seafloor imagery and C.R.G., A.P., S.P.R, E.P.R. and S.B. analyzed the 2019 seafloor imagery. S.B. collected and processed the multibeam bathymetry, and C.R.G., S.S. and S.B. processed and analyzed the sidescan sonar data. C.R.G. drafted an initial manuscript with E.P.R. and W.B. that all authors added further contributions to.

## Competing interests

The authors declare no competing interests.
