## [Peer Review File · Nature Communications]

Volcanically hosted venting with indications of ultramafic influence at Aurora hydrothermal field on Gakkel Ridge.Editorial Note: Parts of this Peer Review File have been redacted as indicated to remove the name of another journal.

REVIEWER COMMENTS

Reviewer #1 (Remarks to the Author):

The manuscript uses observations from plume studies (CH₄-Mn-ratios) and visual observations to suggest the presence of mantle rocks immediately below a thin veneer of basaltic rocks at a neovolcanic axial volcanic ridge. First of all, the involvement of ultramafic rocks in a neovolcanic hydrothermal system on an ultraslow-spreading mid-ocean ridge seems hardly surprising and is probably even more abundant, but the arguments used to come to this conclusion are not convincing. Their conclusion is solely based on 1) the presence of high-temperature black-smoker-type venting at Aurora and 2) the presence of a certain CH₄-Mn-ratio that they claim cannot be explained by sediment interaction.

The argumentation with respect to excluding sedimentary sources for the CH₄-Mn-ratio is, however, weak. The authors claim, for instance, that interaction with thick sediment sequences could produce elevated CH₄-Mn-ratios, but these ratios are supposed to be much higher than the ratio observed at the Aurora vent field. It is not clear to me, however, which sites have been used for this comparison. Is it only data from the Okinawa Trough, because of the continental sediments that are involved? What about other sedimented vent sites, such as the Guaymas and Pescadero basins? Do they show a similar CH₄-Mn-ratio. This is not discussed. What is the influence of a variable sediment thickness on the CH₄-Mn-ratio? The authors further rule out the presence of a major sediment interaction based on the presence of black-smoker-type venting in the first place. This argument is not correct, not even for the Okinawa Trough, the type example used by the authors for another geological setting with terrigenous sediment input. Exit temperatures of over 300°C have been observed at several sites in the Okinawa including Sakai-Noho, Iheya-Aki, Izena Hole, Gondou, and, finally, Yokosuka where temperatures of 364°C have been observed (Miyazaki et al., 2018; Royal Soc Open Sci 4, paper 171570). Water depth rather than the interaction with sediments seems to be the limiting factor for reaching exit temperatures above 300°C for most sites in the Okinawa Trough. Additionally, black smoker-type venting associated with sediments is also known from the Guaymas and Pescadero Basins with some of these vent sites in the southern and northern Troughs reaching exit temperatures of over 330°C. They might be related to thinner sediment cover when compared to the Okinawa Trough, but a discussion of the influence of sediment-thickness and type on the CH₄-Mn-ratio is not provided by the authors.

In summary, an ultramafic influence would not be surprising in this geological setting but has certainly not been proven with this submission. Clearly additional data providing further evidence for the ultramafic involvement is needed. This paper does, therefore, not merit publication in Nature Communications.

Sincerely,

Reviewer #2 (Remarks to the Author):

Dear authors,

the study investigates the geological setting of the high-temperature venting Aurora hydrothermal field on the ultra-slow spreading Gakkel Ridge, which is hardly accessible due to the Arctic ice cover. The study presents near-bottom sidescan sonar data, visual seafloor observations and geochemical plume analyses with the aim to identify the host rock lithology in the subseafloor of the Aurora vent field.

The sidescan data is interpreted for reflectivity patterns, which are linked to seafloor photographs and other ground-truthing observations. Both show the Aurora field to be hosted in highly-reflective, relatively fresh basalt that piles up as a ridge-like volcanic feature that is otherwise surrounded by low-reflectivity, heavily sedimented seafloor. Visual observations reveal a relatively large spatial extent of the field and a variety of venting stages from high-temperature fluids to extinct vent sites. This, together with the ultra-slow spreading nature of the hosting mid-ocean ridge, is used to hypothesise a stable, long-lasting hydrothermal circulation cell underlying the Aurora hydrothermal field. Such stable circulation is typically known from ultramafic sites and contradicts the fresh, basaltic surface lithology observed in the remote sensing data. Therefore, the authors hypothesise an ultramafic lithology in the subsurface of the hydrothermal field underneath a basaltic cover.

An ultramafic host rock signature is identified in the fluid composition of the non-buoyant hydrothermal plume above the hydrothermal field. Analysed He-isotope, dissolved Methane and dissolved Mn concentrations in the plume are assumed to reflect those at the seafloor vent sites. The geochemical ratio of Methane vs. Mn at Aurora compares to ratios known from other ultramafic sites at other ultraslow and slow-spreading mid-ocean ridges.

The study concludes that the Aurora hydrothermal deposit is basalt-hosted at the surface, but the underlying, subsurface circulation cell is dominated by ultramafic host rocks that contribute substantial Methane to the hydrothermal fluid.

Considering the Arctic ice-cover above the Gakkel Ridge, the presented seafloor observations and remote sensing data of the Aurora hydrothermal field are spectacular. The article is well written and the argumentation is good to follow.

The scientific relevance of the characterized geological setting could be better highlighted (see comments below).

I recommend the paper to be published after minor revisions. I split my feedback into some general questions for each section and some line by line comments.

My expertise lies in remote sensing of the seafloor around hydrothermal fields, so my review focusses on the analyses of the sidescan data, seafloor observations, and bathymetric maps.

Best regards,

Meike Klischies

General questions and comments:

Introduction: The introduction needs a sentence explaining, why it is important to better characterize the geological setting of the Aurora hydrothermal field. What gap does the presented findings close and how do they affect our general view on the geological settings of hydrothermal systems?

Also, the study demonstrates clearly that remote sensing alone can only characterize the surficial geological settings, but geochemical analyses of fluids etc. are needed to understand the entire hydrothermal circulation cell including the (deep) subsurface. This should be introduced here.

The history of exploring the Gakkel Ridge (first paragraph) could be shortened.

Results: The extent of the field and the reflectivity patterns in the sidescan image are described in much detail in the text and figure captions. To better illustrate these descriptions, I would wish a (geological) map of the area showing the field's extent and the different acoustic and/or geological entities.

Discussion: To better highlight the significance of the findings, add a sentence about the need for geological characterizations of hydrothermal fields; for example, how do the findings affect the exploration for hydrothermal deposits at other (ultra-slow spreading) mid-ocean ridges.

Figures: The maps in the figures, including the sidescan image, should get a consistent layout (same colour ramp for the bathymetry, coordinate frame, scale bar, north arrow or coordinate grid lines).

Line by line comments:

Line 66: Refer here to the bathymetry in Figure 1.

Line 77: This reads a bit confusing, if you do not know what OFOBS is or is equipped with. Maybe, give a short list of the main instruments of the platform, which would prepare for the following analyses of the sidescan data and photo footage.

Line 80: Is it necessary to mention the failed deployments (OFOBS5) here? Or do you list it, because it provided you with ground truth at distance to the Aurora field? If the latter is correct, rephrase the sentence and state how you used these deployment's data.

Line 91: Could the coring stations be plotted as ground truthing in the Fig.1 or in the sidescan map in Fig.2a?

Line 192: Lucky Strike's Methane:Mn ratio is described here as an exception. Can the ratio of the Aurora plume also be an exception? In the bathymetry, the setting of the Aurora field within the (partly rifted?) Aurora neovolcanic mound appears similar to the setting at Lucky Strike, e.g. Escartín et al. (2014).

Escartín, J. et al. Lucky Strike seamount: Implications for the emplacement and rifting of segment-centered volcanoes at slow spreading mid-ocean ridges. *Geochemistry, Geophysics, Geosystems* 15, 4157–4179 (2014).

Line 206: According to Okino et al. (2015), the Kairei vent field is hosted in basalt at the surface. The fluid chemistry at Kairei is explained with a detachment fault at depth and serpentinization of troctolites (Okino et al., 2015, Nakamura et al., 2009)

Okino, K., Nakamura, K. & Sato, H. Tectonic Background of Four Hydrothermal Fields Along the Central Indian Ridge. in *Subseafloor Biosphere Linked to Hydrothermal Systems* (eds. Ishibashi, J., Okino, K. & Sunamura, M.) 133–146 (Springer Japan, 2015). doi:10.1007/978-4-431-54865-2_11.

Nakamura K, Morishita T, Bach W, Klein F, Hara K, Okino K, Takai K, Kumagai H (2009) Serpentinized troctolites exposed near the Kairei Hydrothermal Field, Central Indian Ridge: Insights into the origin of the Kairei hydrothermal fluid supporting a unique microbial ecosystem. *Earth Planet Sci Lett* 280(1–4):128–136. doi:10.1016/j.epsl.2009.01.024. 21X09000491

Line 225: The argument for an ultramafic subsurface could be strengthened by referring to remote sensing studies of the ultra-slow spreading Southwest Indian Ridge by Sauter et al. (2013), who suggest a thin volcanic cover above

Sauter, D. et al. Continuous exhumation of mantle-derived rocks at the Southwest Indian Ridge for 11 million years. *Nature Geoscience* 6, 314–320 (2013).

Line 323: Add the resolution of the ship-based bathymetry presented.

Line 329: Add the OFOBS survey number for the sidescan sonar data.

Line 338: Explain the dashed lines in Fig.2a in the figure caption.

Figure 1: The legend for the inset's colour scheme is missing. Or, instead, adjust the colours of the inset map to those of the main map. Expand the bathymetry map in Fig.1 to show the regional setting of the Aurora field and neovolcanic mound.

Figure 2: The triangles are too large and hide the ridge-like feature (Fig.2a). This would be solved by presenting the data (maybe including the survey tracks of the other OFOBS tows) and the interpretation separately in two maps. The seafloor photographs (Fig.2c-d) need a scale bar.

Figure 3: I do not understand the benefit for presenting different bathymetry here than in Fig.1. If it is of higher resolution, it should be also used in Fig.1. Add a colour ramp with water depth or adjust the colour ramp and refer to Fig.1's legend in the figure caption.

Figure 4: Label the dashed lines with Lucky Strike and Kairei. Do the lines represent an average? Maybe add the average of the Aurora ratios for better comparison.

Reviewed by Meike Klischies

Reviewer #3 (Remarks to the Author):

The work of German et al in the NCOMMS-22-12440-T ms provides a study of the Aurora hydrothermal vent field along Gakkel Ridge using a series of OFOBS (combined towed HD camera and high-frequency sidescan sonar) surveys and CTD/rosette samples. The authors show images of an active black smoker site at the western most tip of the Aurora Seamount with hydrothermal plume CH₄/dMn (dMn stands for dissolved Manganese) ratios that follows the same trend than the Rainbow hydrothermal plume. They also show evidences for a highly sedimented rift valley, with a sulfide deposit district that may extend laterally over a hundred meters scale. Although the work of German et al does bring new data and information about hydrothermal activity along a ultra-slow spreading ridge, I find that the ms needs improvement before it can be published in Nature Communications. I detail hereafter my main concerns.

Major comments:

Based on the anomaly detected by biogeochemical sensors (temperature and H₂S anomalies), the authors performed OFOBS transects where the seafloor images showed active chimneys with black smoker vents and basaltic pillow lavas. The authors performed towed CTD/rosette casts across the active vent site and argue that the CH₄/dMn ratio observed in the plume falls into the range of ultramafic hydrothermal plumes previously observed (Rainbow, Logatchev and Ashadze). They use this CH₄/dMn ratio close to unity to suggest that the Aurora vent site is under ultramafic influence due to a thin crust cover in the ultra-slow spreading ridge context of the Gakkel ridge.

My main concern with the ms lies in the lack of argument backing up the ultramafic influence hypothesis. Over the 4 examples of ultramafic vent sites known (Rainbow, Logatchev, Ashadze, Kairei) and used by the authors to define the ultramafic range, they discard one of them (the Kairei vent field) without explanation (l.206 – 209). Aurora lies on a neovolcanic basalt mound, and images from the OFOBS show basalts pillows, and smooth thickly-sedimented rift-valley. No dredge samples, no seafloor images nor geophysical data demonstrate that the upper mantle outcrops or is very near the surface. And given the Aurora seamount bathymetry, there is at least 400m (Aurora seamount height) of basalt constituting the upper layer of the crust. Some authors (Cochran et al.2003

<https://doi.org/10.1029/2002JB001830>) show that the crust thickness can be highly variable at Gakkel ridge (e.g. up to 1km east of 32°E to up to 3km west of 32°E to at least 8°E). Further west (i.e. western volcanic zone, 7°W-3°E) - where the Aurora vent site lies - the segment reflects abundant volcanism (Michael et al., 2003 <https://doi.org/10.1038/nature01704>) with a spreading rate around 13.5-14.5mm.yr⁻¹. Additionally, peridotites were not dredged from the Western Volcanic Zone of the Gakkel Ridge (WVZ - Patterson et al., 2021 <https://doi.org/10.1016/j.lithos.2021.106107>). Is there any geological clues that would support a thinner crust (i.e. <1km) at the Aurora mound that would support German et al. interpretation? . Given that hydrothermal convection cells are around 1 to 2 km in diameter, influence of mantle rocks in hydrothermal fluid circulation due to a thin ocean crust cover still appears a bit hypothetical. Were there any other evidences that could back up the authors' hypothesis: significant hydrogen concentrations in the water column for instance?

Minor concern: The authors focused their OFOBS survey around the black smoker site, but from the biogeochemical sensors, other lower, but still significant, temperature and H₂S anomalies were observed during the 2014 cruise (OFOS PS86/065, p66 of the cruise report). How can the authors be sure that there is only a single active high temperature site? None of the 2019 OFOBS surveys were close enough to the aforementioned anomalies to provide images of the seafloor.

Minor comments:

I. 199-201. Although I do agree that dMn won't be that reactive in early hydrothermal plumes, De Angelis et al, 1993 ([https://doi.org/10.1016/0967-0637\(93\)90132-M](https://doi.org/10.1016/0967-0637(93)90132-M)), and Cowen et al, 2002 ([https://doi.org/10.1016/S0016-7037\(02\)00975-4](https://doi.org/10.1016/S0016-7037(02)00975-4)) did show high methane oxidation rates in hydrothermal plume and quick methane removal (1 to 2 km away from the vent site) that severely impacts the CH₄/dMn ratios, and that could for instance lower a high CH₄/dMn ratio in a hydrothermal plume originating from a vent site with a thick sediment cover influence...

I. 206-209. How do the authors explain this Kairei exception?

I. 224-225. The authors quote reference 25 [John, B.E., and Cheadle, M.J., 2010, Deformation and alteration associated with oceanic and continental detachment fault systems: are they similar? Geophysical Monograph v. 188, p. 175-206, 2010.] to argue for a thin ocean crust in ultra-slow spreading ridge context. But ref 25 deals with geotectonical contexts associated with detachment faults, which is not the case here. Maybe another reference would help the authors prove their case better...

Responses to Reviewer Comments

A recurrent theme evident from reading all three reviews is that we tried to fit too much “assumed knowledge” into our original manuscript (~2500 words for [redacted]) and we have now extended the text to ensure that we have allowed enough space to address the reviewers’ key issues. The comments of Rev.2 and especially Rev.3 provided perceptive insights that helped us focus on how best to take advantage of the extra space available in *Nature Communications* - both to expand on critical ideas *and* to add new information, including a new section at the end of the discussion on the broader significance of our work. Below, we present a detailed discussion of the changes that we have made, together with additional supporting comments, in response to each point raised.

Particular changes that we have made to the Discussion, responsive to specific concerns highlighted in the editor’s cover letter, include:

- a) Accentuating the significance of the Aurora site’s size, hence, requisite time-integrated heat flux for formation as an important (but perhaps overlooked) component of our original arguments (Discussion, l.170-199).
- b) Explaining the basis for excluding 2 “outlier” data-points (both Lucky Strike and Kairei are considered anomalous at the global scale, irrespective of this study)
- c) Introducing two new, independent lines of argument to support our hypotheses, one microbiological (l.293-309) and one petrological (l.310-329 & Fig.5)
- d) Adding a new section on the broader significance of our work (also in the abstract) that highlights the potential relevance of our findings; these range from new marine mining opportunities (l.340-360) to helping guide the search for life beyond Earth (l.361-382).

Reviewer #1:

1.1 First of all, the involvement of ultramafic rocks in a neovolcanic hydrothermal system on an ultraslow-spreading mid-ocean ridge seems hardly surprising...

We agree that ultramafic hosted systems *should*, in general, be expected along slower spreading ridges. Nonetheless, what we still consider surprising is that all prior work at Aurora (including two prominent, well-cited *Nature* articles) had predicted the exact opposite for this site. (See also Rev.3’s surprise that our hypotheses appear to be in stark contradiction to “received wisdom” based on pre-existing literature from the same study area). Importantly, our work implies a (globally) novel setting for seafloor venting with wide-reaching implications. We elaborate on this (as encouraged by Rev.2, **point 2.6**) in the revised *Discussion* at **l.331-382**.

1.2 Their conclusion is solely based on 1) the presence of high-temperature black-smoker-type venting at Aurora and 2) the presence of a certain CH₄-Mn-ratio that they claim cannot be explained by sediment interaction.

This reviewer appears to have overlooked a key additional point that informed the hypotheses that we presented – the unusual size and, hence, the large time-integrated heat-flux required to generate the vent field observed at Aurora. Admittedly this was not discussed in much detail within the much shorter original manuscript. While the composition (CH₄/Mn ratio) of the plume argues against simple water-rock reactions with basaltic rocks at Aurora, the size of the field provides a separate and independent argument for why Aurora can no longer be considered a ‘typical’ short-lived neovolcanic-hosted system. We have now sought to make this point more

explicit, including revisions to the abstract (l.29-30, l.33-35) as well as a thorough re-write of the opening section of the Discussion (l.170-199). We have also expanded our title (*Nature Communications* has longer word-limits than [redacted]) to emphasize that what we present here are testable *hypotheses* that could explain our novel and surprising data. Further field-programs would be required for firmer *conclusions* to be reached on the same topic.

1.3 The argumentation with respect to excluding sedimentary sources for the CH₄-Mn-ratio is, however, weak... Do [other sedimented] vent sites show a similar CH₄-Mn-ratio?

In answer to Rev.1's question, MOR vents influenced by *any* kinds of sediment typically exhibit CH₄/Mn ratios that are 1-2 orders of magnitude higher than the values reported here (l.271-272). But we could have done a much better job in the original manuscript explaining that the only reason why we mentioned sediments at all was to make a clear distinction between our results and those from Aurora's nearest known neighbor, Loki's Castle, on the Mohns Ridge. Recognizing that Aurora is also set on an ultra-slow spreading ridge, in the ice-covered Arctic, it is clear that neither back-arc basins (e.g. our own original lines 213-216) nor organic-rich sedimented systems from the Gulf of California (introduced by Rev.1) present viable analogs for Aurora. We do not need to consider CH₄/Mn values to reach that conclusion. In the revised manuscript, therefore, we have chosen to focus upon drawing out a more explicit comparison with Loki's Castle (l.270-292) a site that *does* exhibit clear evidence of terrigenous-sediment-influence. We acknowledge that the Loki's Castle values overlap with values for mid ocean ridge vents overlain by terrigenous sediments but that, because all of those values fall FAR outside the range presented here (Fig.4), we can be confident that sediment influence has no role to play at Aurora.

1.4 Additional data providing further evidence for the ultramafic involvement is needed.

We address this more completely in response to Rev.3 (points 3.2, 3.3). In brief: in addition to (i) the time-integrated heat-flux required to generate the size of the Aurora field and (ii) the characteristic CH₄/Mn ratio of its overlying plume, we now present (iii) evidence from basalt compositions, which are distinct from most of the Gakkel Ridge's Western Volcanic Zone, that are indicative of anomalously thin basaltic crust beneath Aurora (3.2) and (iv) results from a new companion study (now in revision for *Nature Microbiology*) that suggests H₂ release from the Aurora hydrothermal field, consistent with subsurface ultramafic influence (3.3).

Reviewer #2:

2.1 Considering the Arctic ice-cover above the Gakkel Ridge, the presented seafloor observations and remote sensing data of the Aurora hydrothermal field are spectacular.

We appreciate that this reviewer recognized how unique and exciting these data sets are.

2.2 Introduction: The introduction needs a sentence explaining, why it is important to better characterize the geological setting of the Aurora hydrothermal field. What gap does the presented findings close and how do they affect our general view on the geological settings of hydrothermal systems?

Done – see para 2 of revised Introduction (l.59-71)

2.3 Also, the study demonstrates clearly that remote sensing alone can only characterize the surficial geological settings, but geochemical analyses of fluids etc. are needed to

understand the entire hydrothermal circulation cell including the (deep) subsurface. This should be introduced here.

Agreed – but we have deferred any explicit call out of the benefits of our multi-disciplinary approach to the Discussion (e.g. opening of second section, l.202-203) where we can highlight *concrete* examples of the value gained by bringing multiple threads (seafloor surveys, water column chemistry, microbiology, petrology) together to provide novel insights beyond what any one approach could reveal in isolation.

2.4 The history of exploring the Gakkel Ridge (first paragraph) could be shortened.

Done

2.5 Results: I would wish a (geological) map of the area showing the field's extent and the different acoustic and/or geological entities.

Done – see new Fig.2b (geologic interpretation) to match Fig.2a (sidescan survey data)

2.6 Discussion: To better highlight the significance of the findings...

Done – see entire new section added to the Discussion to address this, l.331-382

2.7 The maps in the figures, including the sidescan image, should get a consistent layout

All map figures have been revised along the lines suggested (see detailed comments below)

Rev 2. Line by line comments:

Line 66: Refer here to the bathymetry in Figure 1.

Done – l.52

Line 77: Maybe, give a short list of the main instruments of the platform, which would prepare for the following analyses of the sidescan data and photo footage.

Done – l.83-84

Line 80: Is it necessary to mention the failed deployments (OFOBS5) here?

Good point. OFOBS 05 has been removed, as suggested.

Line 91: Could the coring stations be plotted as ground truthing in Fig.1 or Fig.2?

All coring (2014 expedition) was done before the vent-site was located and lie outside of our sidescan survey area. We have updated the text (l.93-99) to clarify this point and explain, instead, that it is our co-registered video-observations that provide the ground-truthing for the sidescan interpretation.

Line 192: Lucky Strike's Methane:Mn ratio is described here as an exception. Can the ratio of the Aurora plume also be an exception?

Aurora *is* an exception to what has been reported previously from all other vents. But we do not consider Lucky Strike to be a viable analog because Lucky Strike site is **morphologically** anomalous as well as **geochemically** anomalous compared to all other vents. Morphologically, venting at Lucky Strike is hosted atop a 500-1000m tall axial seamount – no comparable feature occurs anywhere else along the ~2500km of survey that first mapped this section of the Mid-Atlantic Ridge (Detrick et al., J.Geophys.Res., 1995) and none are observed on Gakkel Ridge, either (Michael et al., Nature, 2003). Thus, Lucky Strike differs geologically from **all** the many

tens of basalt-hosted vents worldwide that emit fluids with CH₄:Mn ratios < 0.50. The preferred explanation for this is that focused melt extraction and volcanism beneath Lucky Strike must lead to an extremely deep-rooted hydrothermal cell beneath the anomalously *thick* ocean crust - an analogy that does not apply at Aurora. We have amended lines 50-53 of the introduction to make explicit that Aurora lies *deep* within the Gakkel Ridge rift valley (unlike Lucky Strike and precluding arguments for similarities related to anomalously high magmatic focusing / volcanic effusion rates) and expanded the relevant paragraph of the Discussion section (l.214-230) to present the same rationale outlined in this response.

Line 206: According to Okino et al. (2015), the Kairei vent field is hosted in basalt at the surface. The fluid chemistry at Kairei is explained with a detachment fault at depth and serpentinization of troctolites (Okino et al., 2015, Nakamura et al., 2009)

We are grateful to the reviewer for bringing the Okino et al. (2015) paper to our attention. We have included a more complete consideration of the Kairei site in the revised manuscript (l.246-259) as detailed more fully in our response to Rev.3, point 3.1.

Line 225: The argument for an ultramafic subsurface could be strengthened by referring to remote sensing studies of the ultra-slow Southwest Indian Ridge (Sauter et al., 2013).
Done – see revised paragraph opening at l.310-313 and further response to Rev.3, point 3.6.

Line 323: Add the resolution of the ship-based bathymetry presented.
Done

Line 329: Add the OFOBS survey number for the sidescan sonar data.
Done

Line 338: Explain the dashed lines in Fig.2a in the figure caption.
Fig.2 has been updated to include a geologic map with key (Fig.2b)

Figure 1: The legend for the inset's colour scheme is missing...
Color scale added to inset as well as labels to show intersection of Lena Trough/Gakkel Ridge

Figure 2: The triangles are too large... (Fig.2a). This would be solved by presenting the data (maybe including the survey tracks of the other OFOBS tows) and the interpretation separately in two maps. The seafloor photographs (Fig.2c-d) need a scale bar.
All done. Fig.2b provides a geological interpretation to accompany Fig.2a (sidescan data) and track lines for all three OFOBS surveys. Scale bars have been added to each of Figs 2.c,d,e.

Figure 3: I do not understand the benefit for presenting different bathymetry here.
In our original Fig.3a, we presented the bathymetry used to guide the CTD surveys from the same 2014 expedition. We have now replaced that using the same 2019 data as Fig.1.

Figure 4: Label the dashed lines with Lucky Strike and Kairei.
Dashed lines have been labelled.

Reviewer #3:

3.1 My main concern with the ms lies in the lack of argument backing up the ultramafic influence hypothesis. Over the 4 examples of ultramafic vent sites known (Rainbow,

Logatchev, Ashadze, Kairei) and used by the authors to define the ultramafic range, they discard one of them (the Kairei vent field) without explanation (l.206 – 209).

This is an important point (see also Rev.2) and we have expanded this section of the discussion to explain our rationale. Note that there were more than four examples being considered in our original submission. As now described (l.201-292) the global data-set includes *nine* known examples of ultramafic-influenced vent-sites along mid-ocean ridges. Five of those nine sites are high temperature “black smoker” systems located on slow-spreading ridges and, hence, provide the closest plausible geologic analogs to the Aurora site reported here. In the expanded revised version of this section of the Discussion we have added a consideration of the Lost City, Old City and Von Damm hydrothermal fields which we can exclude as candidate analogs for Aurora because (a) they do not host high temperature “black smoker” venting of the type reported here and (b) their CH₄:Mn ratios are off the charts compared to Aurora (CH₄:Mn >> 10:1) – see l.239-246. In the revised text we have also included a more detailed discussion of why we also exclude the Kairei site from our considerations (l.247-259). Just as is the case for Lucky Strike (Rev.2) - Kairei was already recognized as being both geologically anomalous and geochemically anomalous compared to all other known ultramafic ridge-crest vent-systems, prior to our work at Aurora (Okino et al., 2015). Kairei is the only known example of an ultramafic-hosted system on a medium-fast ridge and the only known seafloor hydrothermal system that has low CH₄:Mn ratios in its vent-fluids despite being enriched in both dissolved H₂ and dissolved CH₄. But the focus of our study is to understand the Aurora hydrothermal system on the Gakkel Ridge rather than to attempt to shed new light on why Kairei is anomalous. Having established that Kairei cannot provide a relevant analog setting for Aurora, we do not discuss its own unusual characteristics further, here.

3.2 Aurora lies on a neovolcanic basalt mound, and images from the OFOBS show basalts pillows, and smooth thickly-sedimented rift-valley... Is there any geological clues that would support a thinner crust?

This is another excellent point that we have been happy to address in the revised manuscript. Petrologically, the basalt data-sets from Michael et al. (2003) already provided evidence (from anomalously low CaO/Al₂O₃ ratios) for anomalously low levels of melt generation / basaltic crust thickness at the very westernmost limits of the Gakkel Ridge, where it intersects the Lena Trough. We have taken advantage of both the added word count and number of display items allowable for *Nature Communications*, compared to [redacted], to introduce new Fig.5 to illustrate that point including some confirmatory new analyses of our own. We have revised the Discussion text (l.313-329) to clarify that the ridge-segment hosting the Aurora field, west of 6°W, is petrologically distinct from the remainder of the Western Volcanic Zone (3°E to 5°W).

3.3 Were there any other evidences that could back up the authors' hypothesis: significant hydrogen concentrations in the water column for instance?

Informed by the same pre-existing literature as the rest of our community, we did not anticipate any ultramafic influence at Aurora and, hence, did not anticipate a need to measure for dissolved hydrogen while we were on station in 2014. However, microbial samples were collected and in a complementary study, currently being revised for *Nature Microbiology*, our co-author Antje Boetius' team have isolated a novel member of the genus *Sulfurimonas*, which dominates microbial communities within the Aurora plume samples and exhibits traits indicative of an obligate adaptation to hydrogen rich hydrothermal plumes. Further, the report states that bacterial hydrogenases were highly expressed within the plume, but not above or below it, consistent with H₂ release from the source hydrothermal field. We have added a discussion of these new data and their implications for our work to the revised Discussion at l.293-309.

3.4 How can the authors be sure that there is only a single active high temperature site?

We cannot completely exclude the presence of other hydrothermal sources but as noted in **l.158-163** the coherence of the plume data (see also Figs.3&4) all argue for a single coherent vent-source emanating from Aurora. We have reinforced this point in **l.77-81** by noting that numerous 2014 OFOS tows over the summit of Aurora failed to locate any other active vent-sources. Interestingly, however, the presence of additional active vents would further strengthen our arguments for an unusually large (time-integrated) heat-flux from venting at Aurora, with all sources emitting the same anomalously high CH₄/Mn (± high H₂) vent-fluids.

3.5 Although I do agree that dMn won't be that reactive in early hydrothermal plumes, De Angelis et al, 1993 and Cowen et al, 2002 did show high methane oxidation rates in hydrothermal plume and quick methane removal (1 to 2 km away from the vent site) that severely impacts the CH₄/dMn ratios

Actually, examination of Cowen et al (2002) – notably their Fig.5, panel C - only strengthens our argument. Their two stations close to the ridge axis (Stations 6, 9) define a common trend in CH₄:Mn without fractionation – just as we present here for Aurora (≤3 km range). It is only in their off-axis station (Stn.2) at >5km from the source, that the linear relationship breaks down. Similarly, De Angelis et al (1993) only show attenuation in their dissolved CH₄ concentrations at 20km from the source – much greater than the length scales considered here. In our revised text (**l.260-269**) we acknowledge that biogeochemical transformations in plumes *does* cause fractionation down-stream – but that for the length scales considered here, so close to the source of venting, and with no evidence for fractionation, we do not consider that to be an issue.

3.6. The authors quote reference 25 [John, B.E., and Cheadle, M.J., 2010, Deformation and alteration associated with oceanic and continental detachment fault systems: are they similar? Geophysical Monograph v. 188, p. 175-206, 2010.] to argue for a thin ocean crust in ultra-slow spreading ridge context. But ref 25 deals with geotectonical contexts associated with detachment faults, which is not the case here. Maybe another reference would help the authors prove their case better...

Point well made – see also Rev.2. We have rewritten the relevant paragraph and updated it with more pertinent references specific to ultra-slow spreading ridges (**l.310-313**).

REVIEWERS' COMMENTS

Reviewer #1 (Remarks to the Author):

Review of the paper „ Aurora hydrothermal field, Gakkel Ridge: A volcanically hosted system with indications of potential ultramafic influence “ by German et al.

The manuscript is a resubmission and uses observations from plume studies (CH₄-Mn-ratios) and visual observations to suggest an ultramafic influence on a vent sites located on a basaltic volcanic mound in the high-Arctic, a region very difficult to survey due to its ice cover. The influence is suggested to be caused by the presence of mantle rocks immediately below a thin veneer of basaltic rocks at a neovolcanic axial volcanic ridge. This version of the manuscript has been considerably rewritten and the overall text and especially the discussion have improved considerably. The misleading discussion and comparison with vent sites of the sedimented Okinawa Trough and the wrong statement about the lack of black smoker-style venting in such sedimented systems have been deleted. Instead the comparison is now focused on Lokis Castle. A comparison to the unusual Kairei vent field on the Central Indian Ridge has been included to show the spread of CH₄/Mn ratios globally. The argumentation for an ultramafic influence on the Aurora hydrothermal field is now more stringent due to the changes in the text. Also, further arguments from published petrochemical data have been included that strengthen the authors case for a reduced crustal thickness. The explanation of the importance of the relatively large size of the vent site on the underlying processes has been rewritten and is now much more clear. The authors claim that this site is unusually large when compared to other volcanic-dominated basalt-hosted sites, which is still not entirely true (e.g. Galapagos Rift, Alarcon Rise, and others), but the argumentation that the size of the Aurora vent field is inconsistent with a single dike event and needs long-lived focused fluid flow is correct.

In addition, this revised version of the manuscript now also includes a section highlighting the broader implications of these findings with respect to marine economic geology and even, rightfully, highlighting processes relevant for the search for extraterrestrial life.

Overall, the manuscript improved significantly and many more references were added to strengthen the discussion. Due to these changes, I would therefore recommend publication.

Reviewer #2 (Remarks to the Author):

Dear authors,

The revised manuscript highlights the contradiction between the neo-volcanic setting of the Aurora hydrothermal field, its lateral dimensions indicating sustained, long-lived fluid circulation and its

ultramafic signature in the plume fluids. The sidescan data and visual seafloor observations are used to produce a geological map of the near surrounding of the Aurora hydrothermal field showing the neo-volcanic setting atop a ridge of fresh basalt as well as the (minimum) extent of active venting and (extinct) massive sulphide accumulations that is surprisingly large considering the young volcanic host rocks. The study further presents unusually high CH₄:Mn ratios in the hydrothermal plume that the authors trace back to a deep hydrothermal circulation cell and interaction of the hydrothermal fluid with ultramafic rocks at depth.

The article profits from the larger word count of Nature Communications. The discussion, including the scientific implications of the findings, has drastically improved since the last version and the line of conclusions is good to follow. The only minor concern I have is the studies' impact on the search for extra-terrestrial life as mentioned in the last paragraph of the discussion. I can follow the authors' point that hydrothermal systems like Aurora could accommodate the origin of life on Earth. The comparison of the Gakkel Ridge to Saturn's moon Enceladus could be strengthened, for example, by mentioning not only the ice-cover of Enceladus, but also its tectonic nature (e.g., Kargel, J. S., & Pozio, S., 1996. The Volcanic and Tectonic History of Enceladus. *Icarus*, 119(2), 385–404 <https://doi.org/10.1006/icar.1996.0026>).

I recommend the paper to be published. A few, very minor, line-by-line comments are listed below.

Thank you for addressing the concerns of my co-reviewers and myself so comprehensively.

Best regards,

Meike Klischies

Line by line comments:

Line 218: How about the Menez Gwen segment, North of Lucky Strike, hosting two hydrothermal fields (Menez Gwen and Bubbylon)?

E.g.: Radford-Knoery, J., Charlou, J.-L., Donval, J.-P., Aballéa, M., Fouquet, Y., & Ondréas, H. (1998). Distribution of dissolved sulfide, methane, and manganese near the seafloor at the Lucky Strike (37°17'N) and Menez Gwen (37°50'N) hydrothermal vent sites on the mid-Atlantic Ridge. *Deep Sea*

Research Part I: Oceanographic Research Papers, 45(2–3), 367–386. [https://doi.org/10.1016/S0967-0637\(97\)00082-4](https://doi.org/10.1016/S0967-0637(97)00082-4)

Line 367: Referring to Charles Darwin (Reference 64) is a cool thing to do, but does not play in favour with the seriousness of the discussion - from my point of view. Focus on the second half of the sentence and the results of Reference 11.

Line 370-374: Add here extensional faulting (rift like structures) proposed for Enceladus surface (e.g., Kargel, J. S., & Pozio, S. (1996). The Volcanic and Tectonic History of Enceladus. *Icarus*, 119(2), 385–404. <https://doi.org/10.1006/icar.1996.0026>) to strengthen the similarity to the conditions at Aurora.

Line 524: The figure caption needs to explain the abbreviation “SMZ”.

Line 737: Reference 60 misses the year of publication.

Line 755: A dot is in the wrong position. Instead of “German, C..R” it should be “German, C.R.”

Figure 2b: Remove the question marks from the “sonar shadow” symbology, because the shadows are clearly visible in Fig. 2a - without doubt. The question mark in the western basalt outcrop, however, can remain.

Rev.1

The authors claim that this site is unusually large when compared to other volcanic-dominated basalt-hosted sites, which is still not entirely true (e.g. Galapagos Rift, Alarcon Rise, and others), but the argumentation that the size of the Aurora vent field is inconsistent with a single dike event and needs long-lived focused fluid flow is correct.

An oversight: we had qualified this statement in the text but had not resolved this in the abstract. See l.29-31: *unusually large for a volcanically-hosted vent **on a slow-spreading ridge** and...*

Rev.2

*The comparison of the Gakkel Ridge to Saturn's moon Enceladus could be strengthened, for example, by mentioning not only the ice-cover of Enceladus, but also its tectonic nature (e.g., Kargel, J. S., & Pozio, S., 1996. *The Volcanic and Tectonic History of Enceladus*. *Icarus*, 119(2), 385–404 <https://doi.org/10.1006/icar.1996.0026>).*

We prefer not to introduce this new sub-topic because it is distinct from (hence, might distract the reader away from) the key thrust of this part of our discussion: the potential for habitability / abiotic organic synthesis at the base of Enceladus' ice covered ocean. Kargel and Pozio (1996) anticipated what NASA's Cassini mission would reveal about the exterior ice-shell of Enceladus. But Cassini also revealed unanticipated evidence for a specific form of seafloor hydrothermal venting. Our work at the Aurora field has particular relevance to the latter, but not the former.

Line by line comments:

Line 218: How about the Menez Gwen segment, North of Lucky Strike, hosting two hydrothermal fields (Menez Gwen and Bubblyon)?

The simple answer is that those sites do not host high-temperature (>300°C) "Black Smoker" vent sites, which Aurora demonstrably does. But for completeness' sake, we have revised the manuscript to introduce both sites for their unusually high CH₄:Mn ratios (l.226-227; ref 41) before explaining why they are not viable analogs for the Aurora system (l.228-233; ref 42).

Line 367: Referring to Charles Darwin (Reference 64) is a cool thing to do, but does not play in favour with the seriousness of the discussion - from my point of view.

We have retained the citation because it was not intended facetiously – in Astrobiology two competing hypotheses persist for the origin of life on Earth, (a) deep sea refugia protected during the Late Heavy Bombardment and (b) rapidly drying/rehydrating subaerial environments. The latter remains a rich field of research but it originated with Darwin (ref.66). We have sought to articulate this argument better in revised l.370-390.

*Line 370-374: Add here extensional faulting (rift like structures) proposed for Enceladus surface (e.g., Kargel, J. S., & Pozio, S. (1996). *The Volcanic and Tectonic History of Enceladus*. *Icarus*, 119(2), 385–404. <https://doi.org/10.1006/icar.1996.0026>) to strengthen the similarity to the conditions at Aurora.*

We prefer to maintain a focus exclusively on what this study has to contribute to understanding processes that could be active at the **seafloor** of Enceladus. Most known vents on Earth do not match the conditions observed at Enceladus but one of very few that do (Von Damm, Mid-Cayman Rise) is where we have shown host abiotic organic synthesis to be occurring. Aurora is important in this context as a "rosetta stone" because it is a geologic twin to the Von Damm site but is also situated beneath permanent ice-cover, akin to Enceladus. We have revised and slightly expanded lines 372-385 to refine *this* argument, retaining the relevant citations for Enceladus (12, 65, 67, 68 & 69) in the hope that this clarifies the analog case we seek to make.

Line 524: The figure caption needs to explain the abbreviation "SMZ".
Added to opening sentence of caption.

Line 737: Reference 60 misses the year of publication.
Fixed

Line 755: A dot is in the wrong position. Instead of "German, C..R" it should be "German, C.R."
Fixed

Figure 2b: Remove the question marks from the "sonar shadow" symbology, because the shadows are clearly visible in Fig. 2a - without doubt. The question mark in the western basalt outcrop, however, can remain.

Changed